# Does the Type of Funding Affect Innovation? Evidence from Incubators in China

Chenghua Guan and Shengxue Jin *

School of Economics and Resource Management, Beijing Normal University, Beijing 100875, China
* Correspondence: jinshengxue@mail.bnu.edu.cn

**Abstract:** Technology business incubation is vital for the promotion of innovative development and plays an essential role in economic development and social stability. This paper empirically studies the impact of fund types on incubator innovation and its mechanisms using China's incubator data from 2015 to 2019 and the fixed effect model. It is found that incubation funds, venture capital, and fiscal subsidies can significantly promote incubator innovation, with venture capital having the most substantial boost, followed by incubation funds and fiscal subsidies. Analysis of these mechanisms reveals that the promotion of incubator innovation by different funds relies primarily on R&D expenditure and on the scale of technology services expenditures. Further analysis shows that the effect varies according to the incubator, and that a reduction in the proportion of a comprehensive incubator fund or in the proportion of subsidy for a professional incubator does not contribute to enterprise innovation. This paper provides empirical evidence to support China in its improvement of the financing mechanisms for entrepreneurship and the promotion of sustainable economic and social development.

**Keywords:** incubation innovation; incubation funds; venture capital; fiscal subsidy; funds type; technology services; R&D expenditure

## 1. Introduction

Adequate funding is a fundamental safeguard for incubator technology innovation activities [1]. R&D activities are highly uncertain, and companies tighten their R&D investment and shelve or terminate their R&D activities when there is a lack of financial support. On the contrary, business decision-makers will deregulate investment in risky projects and increase investment in technology research when companies are well-funded, resulting in more robust innovative outputs. To ensure the security of funds, enterprises obtain funds through various financing channels, including bank loans, debt financing, and stock financing.

SMEs have limited financing channels compared with large enterprises. SMEs have little seed capital and low technology maturity, and this can easily lead to moral hazard and reverse selection problems [2,3]. In addition, information asymmetry has existed in the credit market for a long time. Enterprises often strictly control the leakage of trade secrets to maintain a competitive advantage. Investors find it difficult to judge the pros and cons of enterprises, leading to severe financial pressure on SMEs [4,5].

Incubators play an essential role in the innovation economy and are an important part of the national innovation system [6]. As a platform for innovation and entrepreneurship services, the incubator helps early-stage businesses grow by providing space, equipment, management, information, and financial support. This platform has gradually formed a set of operational models integrating space sharing, business consulting, venture capital (VC), mentoring training, and network services [7], significantly contributing to the development of technology-based SMEs. In 2021, R&D expenditure on incubators was $12.9 billion in China, with an average R&D investment intensity of 6.68%. After graduation, the number of listed enterprises reached 6534, with 103 listed on the Sci-Tech innovation board, accounting for one-quarter of the total.

External funding for incubators comes mainly from government subsidies, VC, and incubation funds. R&D, characterized by technology and knowledge, has the spillover characteristics of a public good and is easily imitated by competitors. The private marginal rate of return for enterprise innovation activities is lower than for social activities, stimulating free-riding and weakening the incentive for enterprise-independent innovation [8,9]. Financial subsidies are government incentives to address enterprise innovation's "market failure" [10]. The government encourages enterprises to engage in more innovation activities by providing subsidies to compensate for the loss of benefits in their technological activities [11]. VC and incubation funds fall under the same investment category. The former promotes start-up development through various procedures and instruments that include discreet investigation, evaluation, contract, supervision, direct participation in incubator construction, investment in VC firms, and interaction with VC [12,13]. Incubation funds focus mainly on projects with development prospects that receive external financing through partial equity transfers.

In addition, in the current uncertain situation, some countries are more cautious in fiscal expenditures and more rigorous in project funding reviews. Since 2020, fiscal policy worldwide has generally shifted from unprecedented expansion to tightening, coupled with rising global inflation and tighter financing conditions in 2022, leaving economies with difficult fiscal revenues and expenditures and facing severe fiscal pressures. According to the Institute of International Finance (IIF), global debt increased by $24 trillion to a record $281 trillion in 2021. To alleviate financial pressure and improve the efficiency of funds, some countries, including China, have promoted the refined management of fiscal expenditures and issued many policies calling for the strengthening of project audits and a relentless fight against fraudulent subsidies. The difficulty of applications and the increasing pressure on incubators to increase their value place higher demands on enterprise innovation. At the same time, in the current uncertain economic situation, external investors tend to adopt a relatively conservative investment strategy, paying more attention to the robustness of enterprises and projects [14]. Therefore, it is worth studying how to scientifically allocate external funds, compare the impact of different types of funds on enterprise innovation, and explore the mechanism behind the allocation of external funds scientifically.

Most existing research has focused on the impact of fiscal support, VC, and other funds on corporate performance, fully demonstrating the importance of funding [15]. External funds can alleviate the financing problem of the incubator and improve its performance. In reality, however, incubators have more complex and diverse financing structures than just one funding source. Incubators have different funding sources, with differences in business logic and business objectives behind them, which may lead to an inefficient allocation of funds. When receiving the subsidy, the enterprise must fulfill the target task within the specified time according to the requirements of the government so that the government department can pass the audit. Venture investors obtain proportional equity, forming a community of interest with the enterprise [16]. For sustained value-added profitability, venture investors may participate in enterprise management [17]. Therefore, comparing the benefits of different types of funds can reflect government and market differences. Currently, there are studies that have evaluated the impact of investment portfolios or financial instruments on the performance of enterprises. However, from the perspective of capital type, fewer studies have explored the impact of investment on enterprises [18,19].

In order to make up for the shortcomings of existing research, this paper uses a fixed-effect model based on Chinese incubator data to empirically study the influence of fund type on enterprise innovation and its mechanisms. China is an interesting case study as it has the most significant number of incubators in the world. Since establishing the first technology business incubator at Wuhan's East Lake in 1987, the number of incubators had grown to 6227 by the end of 2021. The development model of the incubator in China is dominated by the government, which guides the development of the industry through fiscal policy and sets up subsidies projects but does not directly intervene in the operation

of the incubator. The incubator industry in China has not yet built a sound operation management system and is in the stage of extensive development. Incubator development suffers from a high dependence on financial subsidies, inefficient use of funds, and low output of innovative results. This paper finds that, in the Chinese context, all capital factors significantly affect enterprise innovation, and that the venture capital promotion effect is the most substantial.

The study's main contributions are as follows. (1) After demonstrating the impact of incubation funds, VC, and financial subsidies on incubator innovation, this study compares the influence of different incubators. We have consolidated the relevant literature on funds and enterprise innovation. From the type of funds, based on validation of the incubator innovation impact of incubator funds, VC, and fiscal subsidies, we have compared the impact of different types of funding to further explore the differences behind funding sources. This paper expands the theoretical research of traditional single funding and provides evidence to enhance the marginal value of funds. (2) The vast majority of research currently targeting incubators has been limited to the use of questionnaire data [20,21] or data at the city level and above [22]. While the former is primarily qualitative research, the latter cannot accurately portray the incubator's microscopic behavior. In addition, related research has defects in data conditions, research methods, and variable selection [23]. This study uses the data from the National Incubator Statistical Database of the Torch Center of the Ministry of Science and Technology, which is extensive and rich in index information. (3) The scale and structure of financing cannot be independently determined by incubators; therefore, it is necessary to reveal the black box of financial activities to gain greater autonomy in allocating funds. This paper analyzes the mechanisms of the different fund types for incubator innovation, providing a reference for the optimization of the allocation of innovative resources and for an improvement in the quality of incubator development.

The structure is as follows. The second part is the literature review, which expounds on the literature research on fund types and incubator innovation. The third part builds the empirical model and explains the variable index. The fourth part starts with the baseline model regression, followed by robustness analysis and endogenous test. The mechanism analysis is carried out in the fifth part to explore the intrinsic influence mechanism of funding type on incubator innovation. The sixth part conducts heterogeneity research from the incubator type. The last part summarizes the study and gives suggestions.

## 2. Literature Review

There is currently less literature that speaks to research on funding portfolios. Different funds differ significantly in investment motivation and value-added services, leading to different development paths for enterprise innovation activities [24]. Some scholars have found that co-investing enterprises show more excellent innovation capabilities than individual firms [25]. Phelps et al. (2012) and Anu Wadhwa (2016) have studied the impact of portfolio diversity on enterprise innovation [26]. Cao and Li (2020) have analyzed the innovative mechanism and effect of portfolio networks built around venture capital institutions [27]. Héctor Cuevas-Vargas. et al. (2022) conducted a questionnaire study on small- and medium-sized manufacturing enterprises in Mexico and found that the capital structure significantly impacted enterprise innovation [28]. Based on transaction costs theory and resource-based theory, Yue and Zhao (2020) and Lan (2023) found that different types of government subsidies and venture capital have utility heterogeneity to enterprise technology [29,30]. In addition, the economic effect represented by government subsidies is related to the internal characteristics and external environment of enterprises [31], which makes the impact of funding sources on innovation activities in different incubators likely to be significantly different. Therefore, we propose the following hypotheses.

**Hypothesis 1.** *The types of funds have a significant impact on enterprise innovation.*

**Hypothesis 2.** *The types of funds have a differentiated effect on enterprise innovation with different characteristics.*

Funding is essential for incubator operations [32]. Incubation activities are characterized by high cost and risk, and adequate capital can alleviate businesses' survival pressure, drive the development, testing, and marketing of innovative products, and increase incubator graduation rates [33]. In addition to self-owned funds, incubators are financed mainly through fiscal subsidies, fund financing, and VC.

Most studies believe that both fiscal subsidies and VC can drive innovation. These act as a sign, sending positive signals to external investors [34,35]. Incubators who receive government support or VC, equivalent to a "guarantee," are more likely to gain recognition from outside investors and can receive more investment and support to enhance incubator innovation [36,37]. In addition, fiscal subsidies provide financial support for innovative activities by increasing revenue [38]. R&D activities are highly uncertain; government subsidies can exert a resource effect, which enables enterprises to improve efficiency and reduce operational risks instead of seeking additional ways to save costs. This relieves financing constraints and cash flow pressures in enterprises, increasing the incentive to innovate by allowing greater profitability. Chirgui et al. (2018) and Guan and Yuan (2018) believe that government supports through fiscal and taxation policies or investment may improve the innovation performance of incubators [39,40].

Compared with financial subsidies, VC directly participates in enterprises' management. It applies industry experience and resources to help enterprises improve governance structures and R&D decisions to enhance innovation value [41,42]. Based on the industry perspective, Kortum and Lerner (2000) studied the impact of VC on the patented inventions of twenty industries in the United States and found that more patents often accompany increased venture capital [43]. Hellmann and Puri (2000) studied 173 high-tech companies in Silicon Valley and concluded that VC stimulates innovation and that VC startups require less time to market their products, increasing the likelihood of the companies going public [44]. Engel and Jerome (2014) proposed a new business innovation model that integrates entrepreneurship, technological change, and VC with fundamental innovations that go beyond the supply chain's technical scope and change management approach of the supply chain [45]. Guo and Jiang (2013) confirmed that VC positively impacts the density of R&D investment, using Chinese manufacturing enterprises as samples [46].

Incubation funds increase their earnings mainly through business value enhancements. Incubation funds are similar to angel investment funds and fall under equity investment. Incubation funds are special funds established by incubators alone or in cooperation to support the development of incubating enterprises. The incubation fund is mainly geared toward early investment and can reduce the innovation risk and operating cost of SMEs and play a role in the innovation process [47]. Compared with VC, incubation funds are relatively small in terms of external innovation network resources. There are three main types of incubation fund: non-differential fund support, project-to-project investment, and direct project investment. The amount of funding these incubation funds offer increases in line with the risks undertaken.

Some scholars have found that financial subsidies also have extrusion effects on business innovation [48,49]. In essence, government subsidy is a form of administrative intervention that increase the market demand for innovation factors, leading to higher factor prices, increased cost of enterprises, and reduced marginal revenue [50,51]. Meanwhile, enterprises may abandon R&D and engage in rent-seeking activities to obtain government subsidies. Under the acts of "seeking subsidies" and "pseudo-R&D," enterprises have adopted strategic innovation measures, and the effective use of government subsidies has dropped significantly [52]. Guan and Chen (2010) analyzed the innovation process of high-tech industries in 26 major cities in China [53]. There are also studies that suggest that there is an inverted "U" relationship between government support, VC, and enterprise innovation rather than a linear relationship. As the scale increases, the fostering effect of fiscal support and VC diminishes

until it becomes an inhibitory effect [54]. Marino et al. (2016) evaluated the effectiveness of public R&D funding policies based on French corporate data. They found that higher funders do not have better R&D innovations than lower funders and non-funders, verifying the existence of inverted "U" relationships from a practical level [55].

Based on the above analysis, the following hypothesis is proposed in this paper.

**Hypothesis 3.** *Different types of funds can promote enterprise innovation, among which the promotion effect of venture capital is more prominent.*

### 3. Model Construction and Variables

*3.1. Model Settings*

A panel linear regression model was constructed in the empirical study. Both univariate and multi-factor test models were used to improve the reliability of the hypothesis testing. The model design is as follows:

$$Y_{it} = \alpha + \beta X_{it} + Z_{it} + \mu_i + \delta_t + \varepsilon_{it} \tag{1}$$

where, $i$ represents the incubator, $t$ denotes the year, $Y_{it}$ is the incubator in the year innovation performance; $X_{it}$ is the funding status for incubators in year $t$; and $Z_{it}$ is a series of control variables. In order to control the influence of factors that do not change with time at the incubator level, regional fixed effect $\mu_i$ and time fixed effect $\delta_t$ are added. $\beta$ is the estimated core parameters, which indicate the incentive effect of different sources of funds on incubator innovation. If positive, it indicates that the financial situation significantly affects incubator performance; however, if it is negative, it indicates that it is not conducive to improving incubator performance. Moreover, $\alpha$ and $\varepsilon_{it}$ are intercept terms versus random error terms, respectively.

*3.2. Samples and Data Sources*

The data are mainly from the "Torch Statistical Survey Information System" database of the Torch Center of the Ministry of Science and Technology in China and the Chinese Urban Statistics Yearbook. As an essential body in formulating and implementing policy related to the incubator, the Torch Center has developed into a banner of science and technology innovation and development in China. In order to guide the high-quality development of the incubator industry, the Torch Centre has conducted an annual survey of incubators and incubators nationwide since 2000, and the resulting data are of great research value. After the sample cleaning, a continuous incubator sample was determined to ensure the validity and reliability of the data, resulting in 971 samples for analysis.

In 2015, the Chinese government formally proposed the initiative of "mass entrepreneurship and innovation", encouraging the development of the incubator and other professional space creation, which led to the rapid development of the incubator industry. The outbreak of COVID-19 in 2020 has posed an unprecedented challenge to the incubator industry in China, and the continued and rigorous prevention of infection has kept incubators from operating on a typical trajectory. In order to eliminate the impact of emergencies such as the "mass entrepreneurship and innovation" initiative and infection prevention, the sample was set for 2015–2019.

*3.3. Description of Variables*

Based on the research of Smilor and Raymond (2013) [56], the number of intellectual property applications was selected to measure incubator innovation. Intellectual property rights are exclusive rights to the achievements created by intellectual labor, including patents, software registration rights, and trademarks. Intellectual property rights are an essential indicator for the evaluation of the innovation capabilities of high-tech enterprises in China. In 2016, China promulgated the Guidelines for the Identification and Management of High-tech Enterprises. This emphasizes that high-tech enterprises must have intellectual property rights. The incremental innovation is fragmented in China, ranging

from traditional manufacturing to high technology. It often takes less than one year to obtain financing to eventually form output [57], so variables are not lagged.

The proportions of incubator funds, VC, and financial aid were selected as the explanatory variables to exclude the scale difference and make a better horizontal comparison. An incubation fund is a form of equity financing similar to angel investment in equity transfers to help start-ups obtain external funding. VC has been closely tied to the innovation of sci-tech enterprises since its inception. The development and growth of high-tech enterprises at home and abroad are inseparable from the support of venture capital. China's VC industry was developed late in 1999, wherein the Chinese government issued *Several Opinions on the Establishment of a VC Mechanism*, formally proposing the basic VC system framework for the first time and kicking off China's venture capital industry. In addition, Local governments attach importance to the creation of a favorable venture capital environment through market-oriented means when formulating public policies. Financial subsidies are a form of prior incentive. China has gradually formed a relatively complete system of financial subsidies for R&D activities since 1986. The primary forms of government subsidies for scientific research include the three types of technology-related costs (trial production of new products, intermediate test fees, subsidies for primary scientific research projects), technological expenses, scientific research infrastructure funding, departmental expenses, defense expenses, enterprise excavation expenses, renovation funds, and other budget funds.

The model added many control variables to eliminate the estimation error caused by missing variables. The average annual population was selected as the proxy variable of a region's population size. Current research has typically used GDP or GDP per capita to measure regional economic development. In order to eliminate the interference of population factors, price-treated GDP per capita was used to measure the regional economic development [58], incubators with a higher level of human capital, and richer innovative thinking and knowledge reserve. The number of practitioners characterizes the incubator scale. In the theory of government support, government support for regional development can provide innovative funds for enterprise development. Therefore, government expenditure for science and technology was controlled in this paper. In addition, all variables are logarithmic after adding 1 to avoid pseudo-regression and eliminate heterogeneity. Descriptive statistics of the data are presented in Table 1.

**Table 1.** Descriptive statistics.

| Type | Symbol | Definition | Mean | Std. Dev. | Min | Max |
|---|---|---|---|---|---|---|
| Interpreted variables | Ipapplication | Applications for intellectual properties | 86.98 | 110.02 | 0 | 575 |
| Explanatory variables | Subsidy | Financial assistance to incubator enterprises (¥1,000,000) | 2.26 | 5.19 | 0 | 37 |
| | Funds | Incubator fund (¥1,000,000) | 18.83 | 48.04 | 0 | 322.8 |
| | VC | Venture capital (¥1,000,000) | 77.25 | 195.37 | 0 | 1357.12 |
| Controlled variables | Personnel | College or above personnel | 825.19 | 712.22 | 4 | 3767 |
| | Employee | Employees in incubating enterprises | 1033.92 | 879.61 | 8 | 4775 |
| | Pergdp | GDP per capita in cities (¥1000) | 97.18 | 40.07 | 24.43 | 184.07 |
| | Population | Average urban population (¥1000) | 765.39 | 475.93 | 117 | 3373.52 |
| | Expenditure | City budget science and technology expenditure (¥1,000,000) | 867.95 | 1239.74 | 4.24 | 4334.17 |
| Other variables | Invention | Number of invention patents | 29.72 | 48.51 | 0 | 288 |
| | Ip | Number of intellectual property rights | 151.14 | 203.92 | 0 | 1187 |
| | R&D | R&D funding (¥1,000,000) | 22.67 | 34.45 | 0 | 195.42 |
| | Service | Revenue from public technology service platforms (¥1,000,000) | 1.17 | 3.81 | 0 | 28.37 |
| | Area | Total area used (1000 m$^2$) | 37.11 | 44.39 | 0.10 | 550 |
| | Revenue | City general public budget revenue (¥1,000,000,000) | 146.97 | 196.66 | 1.40 | 716.51 |
| | Business | Businesses that received financing | 4.03 | 6.90 | 0 | 97 |

## 4. Results

### 4.1. Baseline Model Regression

The unit root and co-integration tests were carried out to avoid pseudo-regression and eliminate the effect of unstable data on regression. In addition, after adopting the F test and the Hausman test, we used the fixed effect model for the empirical study.

We first verified Hypothesis 1, which states that different funds contribute to incubator innovation. The improvement can be compared if the different funds contribute to enterprise innovation. Table 2 shows that the regression coefficients of VC, incubation funds and fiscal subsidies are positive. All three types of funds significantly contribute to incubator innovation, confirming that funds are essential to incubator operation. With the same variable under control, the regression coefficient of VC is greater than that of fiscal subsidy and incubation fund. However, there may be correlations between different types of funds and control variables, which can impact the regression coefficients of the variables, so comparative studies are carried out in the form of proportions.

**Table 2.** Baseline regression results I.

| Variables | ln(Ipapplication+1) | | | | | |
|---|---|---|---|---|---|---|
| | **(1)** | **(2)** | **(3)** | **(4)** | **(5)** | **(6)** |
| ln(Subsidy+1) | 0.035 *** (4.69) | 0.030 *** (4.29) | | | | |
| ln(Funds+1) | | | 0.047 *** (4.83) | 0.039 *** (4.25) | | |
| ln(VC+1) | | | | | 0.071 *** (7.99) | 0.062 *** (7.31) |
| Controls | NO | YES | NO | YES | NO | YES |
| City FE | YES | YES | YES | YES | YES | YES |
| Year FE | YES | YES | YES | YES | YES | YES |
| Observations | 4855 | 4855 | 4855 | 4855 | 4855 | 4855 |
| R-squared | 0.071 | 0.302 | 0.115 | 0.311 | 0.290 | 0.350 |

Notes: Robust standard errors clustered are reported in parentheses. *** indicates statistical significance at the 1% levels.

Table 3 shows that, among the three categories of funds, the regression coefficient of the VC share is positive. In contrast, the coefficient of the fiscal subsidy and incubation fund is negative, and the absolute value of the financial subsidy is more significant than that of the incubator fund. VC contributes more significantly to incubator innovation, followed by the incubator fund and the financial subsidy, which is consistent with Hypotheses 1 and 3. Compared with incubation funds and financial subsidies, VC is more closely integrated with incubators, providing support in management, information, and technology to reduce the cost and risk of incubator operations and drive incubator innovation. Moreover, unlike financial subsidies, the incubation fund investment is more market-oriented, paying more attention to profitability. It mainly invests in enterprises or projects with future development potential. Therefore, these are closely watched by relevant market players, with a relatively high efficiency in allocating funds and more creative output.

In addition, the regression coefficient of the number of people above the tertiary level in enterprises and the urban science and technology budget was significantly positive and passed the significance tests of 1% and 5%, respectively. Human capital is a crucial endowment resource for innovation, one which can strengthen the ability of enterprises to absorb and develop new knowledge. Cities can provide an excellent

ecological environment for enterprise innovation, and fiscal support positively affects incubator innovation.

**Table 3.** Baseline regression result II.

| Variables | ln(Ipapplication+1) | | | | | |
|---|---|---|---|---|---|---|
| | **(1)** | **(2)** | **(3)** | **(4)** | **(5)** | **(6)** |
| Subsidy% | −0.323 ** (−2.75) | −0.294 ** (−2.60) | | | | |
| Funds% | | | −0.242 ** (−2.35) | −0.208 ** (−2.08) | | |
| VC% (Subsidy% = $\frac{Subsidy}{(Subsidy+Funds+VC)}$; Funds% = $\frac{Funds}{(Subsidy+Funds+VC)}$; VC% = $\frac{VC}{(Subsidy+Funds+VC)}$) | | | | | 0.500 *** (4.82) | 0.443 *** (4.46) |
| Personnel | | 0.001 ** (3.12) | | 0.001 ** (2.94) | | 0.001 ** (2.85) |
| Employee | | 0.000 (0.32) | | 0.000 (0.44) | | 0.000 (0.42) |
| ln(Pergdp+1) | | 0.050 (0.36) | | 0.023 (0.16) | | 0.041 (0.29) |
| lnPopulation | | −0.306 (−0.69) | | −0.307 (−0.70) | | −0.325 (−0.74) |
| lnExpenditure | | 0.266 *** (3.64) | | 0.263 *** (3.59) | | 0.262 *** (3.59) |
| City FE | YES | YES | YES | YES | YES | YES |
| Year FE | YES | YES | YES | YES | YES | YES |
| Observations | 4522 | 4522 | 4522 | 4522 | 4522 | 4522 |
| R-squared | 0.077 | 0.123 | 0.075 | 0.122 | 0.085 | 0.129 |

Notes: Robust standard errors clustered are reported in parentheses. *** and ** indicate statistical significance at the 1% and 5% levels, respectively.

### 4.2. Endogenous Testing

Although panel data can address the issue of missing variables, such as individual heterogeneity, the model still suffers from endogenous problems, such as missing variables and reverse causation. Government and external investors are more cautious in choosing subsidies or investment targets, and the funding evaluation system differs. Government subsidies pay attention to the contribution of incubators to the regional economy and employment, while external investments are more concerned with their profitability and growth; however, both involve the investigation of the innovation ability of incubators [12]. Suppose the model contains endogenous explanatory variables, it may also lead to estimator bias, so it is necessary to overcome endogenous problems.

By combining panel data with instrumental variable regression, a fixed-effect model or a first-order difference method can solve the problem of missing variables that do not change over time. The study chose the area of the incubator, city revenue, and the number of enterprises receiving investment and financing as instrumental variables (Hausmann tested the results of instrumental variable regression and original regression). The incubator area is a standard indicator for examining and evaluating incubators (*Measures for the Management of Incubators for Science and Technology Enterprises* clarifies that the declaration of incubators for science and technology enterprises at the national level requires incubators to have an area of not less than 10,000 square meters of self-occupied incubator sites). A city's public revenue is an essential funding source for government projects, which indirectly affects fiscal support for incubators. When municipal public revenues

are small, the government will reduce the scope and size of project subsidies. Enterprises that receive financing have obtained incubation funds, cooperative incubation funds, or external investment, which can reflect the degree to which market investors favor incubators. Regression results were all tested by overidentification and are shown in Table 4 (The regression passes the weak instrumental variable test and the identification test). In addition, GMM allows the random perturbation term in the model to have heterogeneous variance and sequence correlation. GMM was again used in this paper for regression, and the conclusion remains unchanged.

**Table 4.** IV Regression results (Columns (1)–(3) are the result of instrumental variable fixed effect regression; columns (4)–(6) are the results of first-order differential regressions of instrumental variables; columns (7)–(9) are GMM fixed effect regressions; columns (10)–(12) are the results of GMM first-order differential regressions).

| Variables | FE_IV | | | FD_IV | | | FE_GMM | | | FD_GMM | | |
|---|---|---|---|---|---|---|---|---|---|---|---|---|
| | (1) | (2) | (3) | (4) | (5) | (6) | (7) | (8) | (9) | (10) | (11) | (12) |
| Subsidy% | −6.878 *** (−3.97) | | | −10.203 ** (−3.17) | | | −6.878 *** (−3.98) | | | −10.203 ** (−3.17) | | |
| Funds% | | −7.680 *** (−3.80) | | | −5.700 *** (−4.02) | | | −7.680 *** (−3.80) | | | −5.700 *** (−4.02) | |
| VC% | | | 4.125 *** (5.84) | | | 3.792 *** (5.53) | | | 4.125 *** (5.85) | | | 3.792 *** (5.54) |
| Controls | YES | YES | YES | YES | YES | YES | YES | YES | YES | YES | YES | YES |
| Year FE | YES | YES | YES | YES | YES | YES | YES | YES | YES | YES | YES | YES |
| Observations | 4522 | 4522 | 4522 | 3517 | 3517 | 3517 | 4503 | 4503 | 4503 | 3517 | 3517 | 3517 |

Notes: Robust standard errors clustered are reported in parentheses. *** and ** indicate statistical significance at the 1% and 5% levels, respectively.

*4.3. Robustness Test*

4.3.1. Replacing the Core Explanatory Variable

The number of intellectual property applications and the amount of invention patent ownership were used to replace the amount of effective intellectual property ownership. The conclusion is the same as above (See Tables A1 and A2 in Appendix A). The patent application has many advantages in measuring enterprise innovation, and most foreign literature has used patent application quantities to measure innovation. The examination ability of patent-granting institutions seldom constrains the number of invention patent applications in China, which can more objectively reflect the change in innovation capacity over time. According to the Patent Law, invention patents, utility model patents, and design patents are the main types of patents in China. The regression coefficient of incubator innovation was almost unchanged, with the application volume for intellectual property rights and the ownership of invention patents as the core explanatory variables.

4.3.2. Urban Clustering Standard Error Regression

Urban clustering standard error is an exceptionally robust standard error that explains the heterogeneity of clustered cities. Robust standard errors explain the heterogeneity of unexplained variations in the model, considering the circumstances associated with explained variables, usually greater than unsound standard errors. The nature of clustering is the grouping classification of disturbance items, as the disturbance items of the incubator include not only the disturbance factors of the incubator itself but also the disturbance factors of the city. The conclusion remains sound after using urban clustering standard error in fixed effect regression (See Table A3 in Appendix A).

### 4.3.3. Replacing the Data Sample

As Beijing, Tianjin, Shanghai, and Chongqing have extraordinary administrative levels in China, they have superior political resources over the prefecture-level city. Therefore, the four municipalities directly administrated by the central government were removed from the sample. The regression coefficient and significance of incubator innovation performance mostly remained the same (See Table A4 in Appendix A).

## 5. Mechanism Analysis

In the study of government subsidy on enterprise innovation, the focus of debate among scholars is the extrusion effect of government subsidy on individual R&D expenditure. Unlike general enterprises, the mission of incubators is to incubate start-up enterprises. Providing public technical service support is vital to help SMEs innovate. Therefore, this paper analyzes the role of the incubator's innovation from different sources using R&D investment and public technical service as intermediary variables.

### 5.1. Mediation Effect Model Setting

The mechanism of financing structure to incubator innovation from R&D input and technical service is tested in this section. Because all types of funds can promote incubator innovation, financing structure, incubator innovations, R&D inputs, and technical services were integrated into the model regression. We referred to Baron and Kenny's (1986) [59] method to establish the mediation effect model (system of equations) as follows:

$$\ln y_{it} = \beta_0 + \beta_1 x_{it} + \beta_2 \omega_{it} + \lambda_{0i} + \varphi_{0t} + \varepsilon_{0it} \tag{2}$$

$$\ln u_{it} = \gamma_0 + \gamma_1 x_{it} + \gamma_2 \omega_{it} + \lambda_{1i} + \varphi_{1t} + \varepsilon_{1it} \tag{3}$$

$$\ln y_{it} = \alpha_0 + \alpha_1 x_{it} + \alpha_2 \omega_{it} + \alpha_3 \ln u_{it} + \lambda_{2i} + \varphi_{2t} + \varepsilon_{2it} \tag{4}$$

In the above series of equations, Equation (2) indicates the total effect of different channels of funds on the incubator innovation, and the coefficient $\beta_1$ represents the total effect size. Equation (3) reflects the effect of different channels of funds on incubator R&D investment. Among these, $U_{it}$ represents the mechanism variable R&D capital investment and technology platform service, which is the mechanism behind the financing structure for incubator innovation performance which in turn enhances incubator R&D investment and strengthens technology platform service. The coefficients $\alpha_3$ in Equation (4) represent the direct effect of intermediate variables on incubator innovation. Substituting Equation (3) into Equation (4) can further obtain the mediation effect of the intermediary variable on incubator innovation, i.e., the degree to which it maintains an influence through the intermediate conduction of the mechanism variable, which is also a key parameter of concern. The mediator effect model composed of the above three equations portrays the influence mechanism of internal and external financing on incubator innovation.

### 5.2. Empirical Results and Analysis

Due to externalities and high-risk characteristics, R&D activities require sustained financial security. On the one hand, R&D activities have the characteristics of public goods, and innovation outputs will become public goods after the patent protection period. On the other hand, R&D activities are high-risk, a risk that is mainly manifested in the uncertainty of research results and the invisible loss of technical value. This leads to R&D investment in enterprises gaining fewer private benefits than social benefits, resulting in insufficient R&D supply. Incubation funds, VC, and fiscal subsidies can enhance the private benefits of R&D activities, effectively correct the externalities of R&D, motivate start-up R&D, and enhance incubator innovation performance.

VC has more robust oversight and constraints on incubator operation than financial subsidies, considering the funding source. Once an incubator's performance fails to meet investors' needs, VCs can take harsh measures, such as replacing the administrator. This can drive incubators to invest more in R&D; thus, incubator innovation is promoted more substantially. R&D investment can be an intermediary in the capital's size and structure. The capital expansion and the optimization of the capital allocation can enhance the incubator innovation performance by increasing R&D investment.

The public technology service platform is dedicated to meeting the everyday needs of enterprises and is a relatively complete technology innovation support system. To a certain extent, it has the attributes of a public good. It provides professional services such as equipment, facilities, site, consultation, training, demonstration, and technical guidance for enterprises in the cluster to carry out activities such as retrieval, experimentation, testing, and pilot testing. In addition, the public technology services platform provides shared services to incubator in-house businesses. It helps organizations shorten technology R&D and project development cycles, a vital indicator that distinguishes incubators from other innovative and entrepreneurial carriers [60]. In this paper, incubator public technology service platform revenue was selected as the proxy variable of platform technology service. Table 5 reveals that incubators receiving VCs purchased more technical services than incubator funds and have a greater impact on driving incubator innovation. In summary, the type of funding indirectly affects the promotion of incubator innovation through changes in R&D investment and expenditure on public technology services.

**Table 5.** Mediation effect test results.

| Variables | ln(R&D+1) | | | ln(Ipapplication+1) | | | ln(Service+1) | | | ln(Ipapplication+1) | | |
|---|---|---|---|---|---|---|---|---|---|---|---|---|
| | (1) | (2) | (3) | (4) | (5) | (6) | (7) | (8) | (9) | (10) | (11) | (12) |
| Subsidy% | −0.750 ** (−3.01) | | | −0.160 * (−1.72) | | | 0.097 (0.42) | | | −0.298 ** (−2.63) | | |
| Funds% | | −0.095 (−0.46) | | | −0.191 ** (−2.12) | | | −0.565 ** (−2.75) | | | −0.195 * (−1.95) | |
| VC% | | | 0.684 ** (3.11) | | | 0.329 *** (3.85) | | | 0.504 ** (2.42) | | | 0.434 *** (4.35) |
| ln(R&D+1) | | | | 0.790 *** (13.17) | 0.180 *** (13.29) | 0.177 *** (13.08) | | | | | | |
| ln(Service+1) | | | | | | | | | | 0.025 ** (2.66) | 0.023 ** (2.46) | 0.022 ** (2.34) |
| Controls | YES | YES | YES | YES | YES | YES | YES | YES | YES | YES | YES | YES |
| City FE | YES | YES | YES | YES | YES | YES | YES | YES | YES | YES | YES | YES |
| Year FE | YES | YES | YES | YES | YES | YES | YES | YES | YES | YES | YES | YES |
| Observations | 4522 | 4522 | 4522 | 4522 | 4522 | 4522 | 4522 | 4522 | 4522 | 4522 | 4522 | 4522 |
| R-squared | 0.063 | 0.057 | 0.063 | 0.237 | 0.238 | 0.241 | 0.009 | 0.013 | 0.012 | 0.125 | 0.124 | 0.131 |

Notes: Robust standard errors clustered are reported in parentheses. ***, ** and * indicate statistical significance at the 1%, 5% and 10% levels, respectively.

## 6. Further Research

There are two types of incubators, the comprehensive incubator and the professional incubator. Comprehensive incubators accept new start-ups with development potential in any industry and pursue rapid growth. In recent years, these have paid particular attention to establishing or utilizing the park's intermediary network service to allocate essential resources such as VC to develop enterprises. The professional incubator focuses on incubating enterprises in particular technical areas and requires a minimum industry concentration of 75%. In addition to its general function as a comprehensive incubator, the professional incubator also has the function of establishing an online technology platform for enterprises and providing standard technical equipment and professional technical support.

The results are shown in Tables 6 and 7. For comprehensive incubators, the proportion of incubator funds had no significant impact on incubator innovation. Financial subsidies outperform VC in driving R&D investment, resulting in relatively weak incubator innovation. Reducing the proportion of subsidies boosts enterprise R & D investment, thus promoting enterprise innovation. For professional incubators, VCs can drive public technical service expenditure and promote incubator innovation more significantly than incubator funds. Meanwhile, the proportion of financial subsidies for incubator innovation is negative but insignificant, and the conclusion is in line with Hypothesis 2. The reduction in the share of incubation funds can increase the size of technical services and accelerate the process of technological innovation.

**Table 6.** Heterogeneity analysis results.

| Variables | Comprehensive Incubator | | | Professional Incubator | | |
|---|---|---|---|---|---|---|
| | ln(Ipapplication+1) | | | | | |
| | (1) | (2) | (3) | (4) | (5) | (6) |
| Subsidy% | −0.291 ** (−2.09) | | | −0.217 (−1.06) | | |
| Funds% | | −0.206 (−1.64) | | | −0.276 * (−1.66) | |
| VC% | | | 0.441 *** (3.64) | | | 0.430 ** (2.34) |
| Controls | YES | YES | YES | YES | YES | YES |
| City FE | YES | YES | YES | YES | YES | YES |
| Year FE | YES | YES | YES | YES | YES | YES |
| Observations | 3106 | 3106 | 3106 | 1416 | 1416 | 1416 |
| R-squared | 0.109 | 0.108 | 0.115 | 0.163 | 0.166 | 0.171 |

Notes: Robust standard errors clustered are reported in parentheses. ***, **, and * indicate statistical significance at the 1%, 5% and 10% levels, respectively.

The results are shown in Tables 6 and 7. For comprehensive incubators, the proportion of incubator funds had no significant impact on incubator innovation. Financial subsidies outperform VC in driving R&D investment, resulting in relatively weak incubator innovation. Appropriately reducing the proportion of financial subsidies can boost enterprise R&D investment and thus promote enterprise innovation. For professional incubators, VCs can drive public technical service expenditure and promote incubator innovation more significantly than incubator funds. Meanwhile, the proportion of financial subsidies for incubator innovation is negative but insignificant; a conclusion which is in line with Hypothesis 2. The reduction in the share of incubation funds can increase the size of the technical services and accelerate the process of technological innovation.

**Table 7.** Results of heterogeneous mechanism analysis.

| Variables | Comprehensive Incubator | | | | | | Professional Incubator | | | | | |
|---|---|---|---|---|---|---|---|---|---|---|---|---|
| | ln(Ipapplication+1) | ln(R&D+1) | ln(Ipapplication+1) | ln(Ipapplication+1) | ln(R&D+1) | ln(Ipapplication+1) | ln(Ipapplication+1) | ln(Service+1) | ln(Ipapplication+1) | ln(Ipapplication+1) | ln(Service+1) | ln(Ipapplication+1) |
| | (1) | (2) | (3) | (4) | (5) | (6) | (7) | (8) | (9) | (10) | (11) | (12) |
| Subsidy% | −0.291 ** (−2.09) | −0.519 ** (−2.00) | −0.197 (−1.59) | | | | | | | | | |
| Fund% | | | | | | | −0.276 * (−1.66) | −0.496 (−1.46) | −0.282 * (−1.68) | | | |
| VC% | | | | 0.441 *** (3.64) | 0.445 * (1.68) | 0.361 *** (3.39) | | | | 0.430 ** (2.34) | 0.684 * (1.81) | 0.440 ** (2.37) |
| ln(R&D+1) | | | 0.180 *** (10.71) | | | 0.179 *** (10.73) | | | | | | |
| ln(Service+1) | | | | | | | | | −0.012 (−0.73) | | | −0.014 (−0.85) |
| Controls | YES | YES | YES | YES | YES | YES | YES | YES | YES | YES | YES | YES |
| City FE | YES | YES | YES | YES | YES | YES | YES | YES | YES | YES | YES | YES |
| Year FE | YES | YES | YES | YES | YES | YES | YES | YES | YES | YES | YES | YES |
| Observations | 3106 | 3106 | 3106 | 3106 | 3106 | 3106 | 1416 | 1416 | 1416 | 1416 | 1416 | 1416 |
| R-squared | 0.109 | 0.062 | 0.223 | 0.115 | 0.062 | 0.228 | 0.166 | 0.013 | 0.166 | 0.171 | 0.015 | 0.172 |

Notes: Robust standard errors clustered are reported in parentheses. ***, **, and * indicate statistical significance at the 1%, 5% and 10% levels, respectively.

## 7. Discussion

### 7.1. Conclusions

The abundance of innovative funding and the availability of financing channels determine the sustainability of innovation. Using data from a Chinese incubator, we compared the influence of capital type on enterprise innovation and analyzed its mechanisms. We have found that incubator funds, incubating funds, and VC significantly facilitate incubator innovation. The promotion of VC is the most substantial of them all. The difference in the effect on incubator innovation is indirectly caused by different funds, mainly through R&D expenditure and technical services support. In addition, from the incubator type, the influence of capital type on incubator innovation is heterogeneous, and the promotion of incubator innovation by integrated incubator financial subsidy is relatively weak.

### 7.2. Implication for Theory

This study integrates the literature on capital and innovation and raises the understanding of resource endowment to enterprise innovation. Existing research fully demonstrates the importance of funding, but the growth of enterprises is independent of a specific type of funding. The multi-fund investment enables the enterprise to have a "pool" of funds and also provides conditions for the enterprise to allocate funds. We have ostensibly studied the impact of funding types, but the allocation of kernel-focused factor resources is related to the combination of funding and other elements. The study's conclusion shows that enterprises will differentiate different types of funds. If R&D investment can be promoted or the supply of technical services increases, enterprise innovation will significantly improve. In addition, venture capital and incubation funds come from the market and from government subsidies. This study has explored how markets and governments work together in enterprise innovation from a financial perspective. Compared with financial subsidies, venture capital and incubation funds can promote enterprise innovation and provide essential references for market theory and regional innovation ecological research.

In addition, many scholars have demonstrated the importance of incubators in fostering SMEs, especially emphasizing the role of incubators in building incubation networks to improve their performance. This study can also be classified as social innovation network research, selecting traditional capital elements as the entry point, exploring the capital network of incubating enterprises, and enriching incubator management theory.

### 7.3. Implication for Practice

More funds are needed to develop enterprises, especially for technology-based SMEs. For industrialized countries, technology-based small- and medium-sized enterprises (SMEs) are the driving force of economic growth and the vital source of budget tax. Research on the performance of different types of funds can help SMEs that lack funds clarify the elements' differentiated value and make the most reasonable choice under financing constraints.

The conclusion of this study has some guiding effect on Incubator operation. Incubators are the intermediary services that are closest to start-ups and are also active in helping them to grow and to provide the right solutions for business financing. On the one hand, incubators promote government subsidy policies for enterprises and give handheld guidance when applying. On the other hand, the incubator connects with outside investors and invites them to participate in a roadshow for the incubator business, doing everything possible to build channels for both parties [61]. Therefore, incubators should weigh the relationship between financial subsidies and external investment, enhance their service concepts, and help enterprises improve innovation efficiency.

*7.4. Suggestions*

In order to promote the high-quality development of incubators, we should give full play to the role of government regulation and market mechanisms so that the two can work together. The government should be deeply aware of the critical role of incubators in innovative entrepreneurial activities, consider incubators as potential economic growth points and technological breakthroughs, and increase incubator support. Under tightening monetary funds, we can appropriately reduce subsidies and strengthen capital market construction, mainly venture capital. If the incubators combine with VC, funds, and technology investment, the capital will better serve the long-term development of enterprises. In addition, the capital market plays a full role in enterprise innovation, promotes the market-oriented allocation of innovation elements, expands incubation networks, and strengthens the entire flow of technology, management, talent, and other factors. At the same time, we should strengthen the public technology service platform, enhance the service capacity and knowledge level of technicians, create a sound innovation ecology for the development of enterprise, and realize innovation-driven sustainable development.

*7.5. Limitations*

There are still many limitations in this study. Limited by the data, we selected quantifiable R&D expenditure and technical service indicators to explore their intermediary effect on enterprise innovation. However, the process from capital input to innovation output needs to go through multiple links, and the mechanism of fund function is complex. At the same time, we have focused only on the innovative benefits of funding while ignoring its costs and possible adverse effects, which require more detailed microdata support. In addition, regarding the measurement of enterprise innovation, intellectual property rights and invention patents were selected as agency indexes, but for different investors, their focus on innovation is different. Governments pay attention to the type and quantity of patents. At the same time, outside investors are more concerned about the commercialization prospects of patented technologies, preferring technologies that can be quickly transformed into products. The current research fails to portray the specific effect of capital type on enterprise innovation, which is also an important research direction to reveal the effect of funds.

**Author Contributions:** Data curation and draft, S.J.; methodology, review and editing, C.G. All authors have read and agreed to the published version of the manuscript.

**Funding:** This research was funded by the National Natural Science Foundation of China (Study on the Influence of Social Capital and Preferences on Environmental Protection Behavior; Project NO: 71773010).

**Institutional Review Board Statement:** Not applicable.

**Informed Consent Statement:** Not applicable.

**Data Availability Statement:** Not applicable.

**Acknowledgments:** The authors thank the editor and the anonymous reviewers for their useful comments and suggestions.

**Conflicts of Interest:** The authors declare no conflict of interest.

# Appendix A

**Table A1.** Robustness test results: changing the explanatory variables I.

| Variables | ln(Invention+1) | | | | | |
|---|---|---|---|---|---|---|
| | **(1)** | **(2)** | **(3)** | **(4)** | **(5)** | **(6)** |
| Subsidy% | −0.179 * (−1.80) | −0.147 (−1.54) | | | | |
| Funds% | | | −0.203 ** (−2.24) | −0.175 ** (−2.00) | | |
| VC% | | | | | 0.348 *** (3.47) | 0.295 ** (3.04) |
| Controls | NO | YES | NO | YES | NO | YES |
| City FE | YES | YES | YES | YES | YES | YES |
| Year FE | YES | YES | YES | YES | YES | YES |
| Observations | 4522 | 4522 | 4522 | 4522 | 4522 | 4522 |
| R-squared | 0.051 | 0.098 | 0.052 | 0.098 | 0.056 | 0.101 |

Notes: Robust standard errors clustered are reported in parentheses. ***, ** and * indicate statistical significance at the 1%, 5% and 10% levels, respectively.

**Table A2.** Robustness test results: changing the explanatory variables II.

| Variables | ln(Ip+1) | | | | | |
|---|---|---|---|---|---|---|
| | **(1)** | **(2)** | **(3)** | **(4)** | **(5)** | **(6)** |
| Subsidy% | −0.225 * (−1.95) | −0.179 * (−1.66) | | | | |
| Funds% | | | −0.234 ** (−2.28) | −0.196 (0.00) | | |
| VC% | | | | | 0.415 *** (3.73) | 0.340 ** (3.28) |
| Controls | NO | YES | NO | YES | NO | YES |
| City FE | YES | YES | YES | YES | YES | YES |
| Year FE | YES | YES | YES | YES | YES | YES |
| Observations | 4522 | 4522 | 4522 | 4522 | 4522 | 4522 |
| R-squared | 0.153 | 0.230 | 0.153 | 0.231 | 0.159 | 0.234 |

Notes: Robust standard errors clustered are reported in parentheses. ***, ** and * indicate statistical significance at the 1%, 5% and 10% levels, respectively.

**Table A3.** Robustness test results: urban clustering standard error analysis.

| Variables | ln(Ipapplication+1) | | | | | |
|---|---|---|---|---|---|---|
| | **(1)** | **(2)** | **(3)** | **(4)** | **(5)** | **(6)** |
| Subsidy% | −0.323 ** (−2.75) | −0.309 ** (−2.73) | | | | |
| Funds% | | | −0.242 ** (−2.35) | −0.210 ** (−2.09) | | |
| VC% | | | | | 0.500 *** (4.82) | 0.457 *** (4.58) |
| Controls | YES | YES | YES | YES | YES | YES |
| City FE | YES | YES | YES | YES | YES | YES |
| Year FE | YES | YES | YES | YES | YES | YES |
| Observations | 4522 | 4522 | 4522 | 4522 | 4522 | 4522 |
| R-squared | 0.077 | 0.123 | 0.075 | 0.122 | 0.085 | 0.130 |

Notes: Robust standard errors clustered are reported in parentheses. *** and ** indicate statistical significance at the 1% and 5% levels, respectively.

**Table A4.** Robustness test results: replacing the data sample.

| Variables | ln(Ipapplication+1) | | | | | |
|---|---|---|---|---|---|---|
| | (1) | (2) | (3) | (4) | (5) | (6) |
| Subsidy% | −0.221 * (−1.78) | −0.200 * (−1.65) | | | | |
| Funds% | | | −0.218 * (−1.89) | −0.195 ** (−1.74) | | |
| VC% | | | | | 0.427 *** (3.58) | 0.383 *** (3.34) |
| Controls | NO | YES | NO | YES | NO | YES |
| City FE | YES | YES | YES | YES | YES | YES |
| Year FE | YES | YES | YES | YES | YES | YES |
| Observations | 3669 | 3669 | 3669 | 3669 | 3669 | 3669 |
| R-squared | 0.074 | 0.114 | 0.075 | 0.114 | 0.081 | 0.119 |

Notes: Robust standard errors clustered are reported in parentheses. ***, ** and * indicate statistical significance at the 1%, 5% and 10% levels, respectively.

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
