# Peer review of "Does the Type of Funding Affect Innovation? Evidence from Incubators in China"

_sustainability, doi:10.3390/su15032548_

Round 1

Reviewer 1 Report

 Review comments

I share my major concerns, which should be incorporated before the paper is accepted.

Major comments:

1.     The study needs to provide contribution points in introduction section and logics. How different type of financing effect the types of incubator? And what is the theoritical reasoning behind this?

2.     The author need to indicate the theoritical importance of the study, in introduction section.

3.     Page 2 – Line 45 – Provide the reference for “Start-ups prefer innovation com-45 pared to large companies but have less seed capital and low technology maturity, which 46 can easily lead to ethical hazards and reverse selection problems. In addition, information 47 asymmetry persists in the credit market.”

4.     Page 2 – Line 66-67. Explain this in details “Some countries have become more cautious in fiscal expenditure, 66 more rigorous project funding review, increasing difficulty in applying, putting more 67 pressure on incubators to add value, and placing higher demands for innovation.”

5.     Literature is missing, I recommend that authors have to use include the recent literature.

6.     Studies on the related topic are heavily available. The econometric equation, estimation models are all old norms. The analysis is too simple. Moreover, the authors just present the test results but fail to further discuss the results. In addition, there are many syntax and formatting errors. In total, the authors need to explain fully why such a study is significant enough to be published in an important journal such as Sustainability. Furthermore, as per the comment to show the significance of the study, the article incorporated:

Reviewer 2 Report

We found that your article is quite good and shows new things, in the development of innovation and sustainability based on the type of funding, but we submit some corrections as follows:

1. What does mean point 4. Mechanism Analysis? (see line 313), is this part of Results or does this point 5 mean?

2. It is necessary to explain the theory or references related to the Comprehensive you mentioned in the discussion chapter.

3. For the concluding chapter, points 1, 2, 3 and 4 do not need to be made, it's better to just narrate it directly or just make it in one paragraph.

Reviewer 3 Report

Dear Authors,

Below you can find my comments on your paper:

1. The finding shared in the abstract (last 4 lines) are not clear at all. Try to share exactly what you have found or contributed to the research field.

2. Many more relevant keywords can be added to the paper.

3. In introduction section, you can explain a bit more about the research gap. Those 3-4 sentences are not well-supported and enough to show the gap.

4. In literature review many of the references are not very recent. Having more up to date references shows that the researchers have worked on this section properly.

5. The justification at the end of page 4, is not strong enough for selecting years 2015-19 as your sample. Try to either improve your justification or the data range.

6. On page 6, why you have collected data about the "area" variable? and do you think a business can run in 0.1 square meter?

7. title of section 4.1 is Baseline not Daseline.

8. Discussion and conclusion sections are too short and shallow. You have a wide range of finding in your study but their interpretation, comparing with previous research and sharing their implications on incubator innovation is almost not shared.

9. As shared above, beside the weak justification to do the analysis on 2015 to 2019, the paper looks like a paper written 4 years ago on year 2019, as you have almost no citation to the studies after 2018 which is a critical issue and needs careful attention.

Reviewer 4 Report

Here are the suggestions before further consideration.

1. Do the incubators in China include both foreigners-founded startups?

2. The authors should mention and conclude the test of assumptions of linear regression to ensure that the analysis is valid.

3. The authors should conclude the test of three hypotheses, formulated in Section 2.

4. Errors in heading number Lines 307 and 313.

5. The authors should provide theoretical and practical contributions from the findings.

6. Can the authors provide suggestions for further research?

Round 2

Reviewer 1 Report

The author has addressed all the comments

Reviewer 3 Report

Dear Authors,

All my concerns are met properly and the updates version of the paper looks fine.

Reviewer 4 Report

Congratulations!